# A Machine-Learning-Based Approach to Predict Deforestation Related to Oil Palm: Conceptual Framework and Experimental Evaluation

**Tarek Sboui** [1,2,*] **, Salwa Saidi** [1,3] **and Ahmed Lakti** [1]

1   Department of Geology, Faculty of Science of Tunis, University of Tunis Al-Manar, Tunis 1068, Tunisia
2   GREEN-TEAM Laboratory, INAT, Tunis 1082, Tunisia
3   Water, Energy and Environment Laboratory (LR3E), ENIS.Bpw 1172, Sfax 3038, Tunisia
*   Correspondence: tarek.sboui@fst.utm.tn

**Featured Application: This work applies machine learning to enhance the prediction of deforestation related to oil palm. This research can be used for decision makers trying to foresee and manage deforestation caused by palm oil production. Providing information about deforestation prediction can help users to make appropriate decisions about where and when they can establish new plantations to ensure a sustainable oil palm production.**

**Abstract:** Deforestation is recognized as an issue that has negative effects on the ecosystem. Predicting deforestation and defining the causes of deforestation is an important process that could help monitor and prevent deforestation. Deforestation prediction has been boosted by recent advances in geospatial technologies and applications, especially remote sensing technologies and machine learning techniques. This paper highlights the issue of predicting deforestation related to oil palm, which has not been focused on in existing research studies. The paper proposes an approach that aims to enhance the prediction of deforestation related to oil palm plantations and palm oil production. The proposed approach is based on a conceptual framework and an assessment of a set of criteria related to such deforestation. The criteria are assessed and validated based on a sensitivity analysis. The framework is based on machine learning and image processing techniques. It consists of three main steps, which are data preparation, model training, and validation. The framework is implemented in a case study in the Aceh province of Indonesia to show the feasibility of our proposed approach in predicting deforestation related to oil palm. The implementation of the proposed approach shows an acceptable accuracy for predicting deforestation.

**Keywords:** machine learning; deforestation prediction; accuracy; criteria assessment; image processing

## 1. Introduction

Deforestation is the loss of tree cover over a given time period, usually as a result of forests being cleared for other land uses, such as farming or ranching [1]. Over the past years, there has been a large amount of deforestation, which has affected the ecosystem. Deforestation is recognized as an issue that has negative effects in the ecosystem. There are many cases where people deforested areas without following rules and regulations [2]. Between 2015 and 2020, the rate of deforestation was estimated to be 10 million hectares per year [3]. Deforestation is caused by human and natural factors. Natural factors include forest fires and forest tree diseases. On the other hand, human activities are among the main causes of global deforestation. In fact, deforestation usually occurs when forests are cleared for human use, such as creating new houses and establishing new plantations. According to the Food and Agriculture Organization (FAO), the expansion of agricultural areas has caused nearly 80% of global deforestation. The remaining causes of deforestation

are diverse, including the construction of infrastructures, such as roads or dams, and mining activities and urbanization [4].

Due to the accelerated deforestation in the last decades, and considering the importance of forest and its impact on the global environment, deforestation needs to be analyzed and compensated in order to reduce its negative effects on the ecosystem and to promote sustainable development [5].

One of the aspects of deforestation is the production process of palm oil, which is an edible vegetable oil that comes from the fruit of oil palm trees [6]. Palm oil production can lead to the clearing of new land to create more space for plantations or to establish new plantations. One major issue is that palm oil production is not easy to oversee, as large areas are cultivated by smallholders. Nongovernmental institutions, such as the Roundtable on Sustainable Palm Oil (RSPO), are trying to tackle these problems and aim to ensure a sustainable palm oil production [7].

In addition, we have witnessed a growing availability of multisource forest data due to new remote sensing methods and techniques [8]. Several studies have demonstrated the usefulness of remote sensing images as an information source for forest management [9,10].

Forest satellite images have been processed using machine learning (ML) techniques [11–13]. Machine learning is a form of artificial intelligence, in which a computer is algorithmically trained to perform a task, such as event prediction or image classification [14,15]. The advantages of machine learning include flexibility and scalability compared with traditional statistical techniques, which makes it deployable for many tasks, such as phenomena stratification/clustering, classification, and predictions. Another advantage of machine learning algorithms is the ability to analyze diverse data types (e.g., geospatial data and descriptive data) and incorporate them into predictions [16]. Recent works show that machine learning techniques have proved to be an effective solution to detect deforestation [17–19].

While several works have shown promising results in detecting deforestation related to oil palm, there has been no research study that focuses on the prediction aspect of deforestation related to oil palm (i.e., oil palm plantations and palm oil production).

Additionally, with regard to the application of machine learning in preventing deforestation, existing works have not assessed the criteria considered in model training. Consequently, there is still uncertainty about the effect of each criterion in the deforestation prediction.

This research study focuses on the prediction aspect of deforestation related to oil palm. It aims to reduce uncertainty about the effect of each criterion by evaluating the influence of each criterion in the prediction of oil palm deforestation.

The main contribution of our approach relies on focusing on the prediction of deforestation related to oil palm. We developed a deforestation prediction model that can serve as useful information for decision makers trying to map and monitor deforestation caused by oil palm plantations and palm oil production.

Another contribution of our work is the evaluation of the effect of each criterion in predicting deforestation related to oil palm. Such evaluation would enhance the performance of model training by focusing on a small set of criteria (i.e., the most important criteria). Multicriteria evaluation is based on a sensitivity analysis that relies on changing only one criterion at a time while keeping the others constant.

The novelty of this study relies on the combination of different aspects: the prediction, the sensitivity analysis, and the use of machine learning in order to enhance the accuracy of the model results.

In the next section, we review some proposed approaches to detect deforestation based on machine learning. In Section 3, we present our proposed approach to enhance the prediction of deforestation related to oil palm. Our approach consists of a set of criteria for detecting deforestation and a framework based on machine learning that aims to detect deforestation. Sensitivity analysis is used to validate weights of criteria. In Section 4, we present a prototype developed to implement our approach. In Section 5, we conclude the study and present some perspectives for future study.

## 2. Related Works

The feasibility of machine learning approaches has been demonstrated in applications, such as earth observation [20], detecting changes on the earth's surface [21], and fire management [22].

Several studies have used machine learning for forest management and deforestation [23]. Studies have focused on various applications, including estimation of forest biophysical properties based on decoders and encoders [24,25], plant pattern identification and classification [26–32], semantic segmentation [33], and assessment of the sustainability of forest management [9,10,34]. Additionally, machine learning studies have been carried out on the interpretation and extraction of forest features [35–38].

With regard to deforestation, which is the general scope of this paper, machine learning approaches can be classified into two categories: approaches detecting the location of areas at risk of deforestation and approaches analyzing the variables that drive deforestation [39]. Chang et al. [40] proposed a machine learning model to enhance the estimates of forest land cover type and forest structural metrics. It is a multitask model that performs both classification and regression concurrently, thereby consolidating several independent tasks and models into one stream. Maeda et al. [41] applied a machine learning model to detect land use changes in the Amazon. Based on change interpretation, they could identify areas with high risk of being burned and improve current fire scar mapping by enabling the distinction between fires in primary forests and fires in previously burned areas. Kehl et al. [42] proposed a study to detect daily deforestation in the Amazon rainforest. They developed an approach to train machine learning models on satellite images, and conducted a spectrum temporal analysis of the deforestation area. The approach aided in understanding the dynamics of the deforestation in the Amazon rainforest. Rosa et al. proposed a spatially explicit model of deforestation in order to predict the potential magnitude and spatial pattern of Amazon deforestation. The model was validated and identified spatial areas of deforestation that accumulates over time [43]. Ye et al. developed a spatially explicit model for detecting Australia's forest cover change using long short-term memory (LSTM) deep learning neural networks. The model was applied to a multidimensional spatiotemporal dataset. The results showed that the model is reliable for projecting forest cover and agricultural production [44].

With regard to palm oil, some studies have focused on the detection of oil palm plantations based on satellite imagery. Li et al. used the texture of trees to train machine learning models in order to distinguish oil palm from high-resolution remote sensing images of Malaysian forests. The implementation results showed that most of the oil palm trees were correctly detected [45,46].

Cheng et al. used a support vector machine (SVM) classifier and a Mahalanobis distance (MD) classifier to undertake supervised classifications of oil palm plantations. They used Landsat, PALSAR, and combined Landsat and PALSAR data for two locations in peninsular Malaysia. Results indicate that accuracies from combined Landsat and PALSAR are better than accuracies from Landsat or PALSAR alone for both study areas using both classifiers [47]. Xu et al. [48] proposed a method to improve the accuracy of detecting and classifying oil palm types (e.g., mature or young). They used the random forest algorithm based on improved grid search optimization (IGSO-RF) in order to establish a classification model and detect oil palm plantations. The results showed that the proposed method improved the detection accuracy of oil palm plantation.

While the aforementioned machine-learning-based works showed promising results in detecting deforestation and in identifying the extent of oil palm areas, some challenges still need to be addressed, including the inadequate samples of imagery and the limitation of the number of parameters considered in the training model. Additionally, existing works have not focused on predicting deforestation caused by palm oil production and oil palm plantation.

In this work, we focus on predicting deforestation related to oil palm. We aim to enhance the accuracy of the prediction based on machine learning. For that, we propose

an approach that enriches the existing commonly used parameters for machine learning. Additionally, we perform sensitivity analysis to assess the effect of each parameter affecting oil palm deforestation.

The particularity of our approach lies in the prediction of deforestation related to palm oil. The developed prediction can serve as a useful document for decision makers trying to foresee and manage palm oil deforestation. Another particularity of our work is the validation of the machine training via sensitivity analysis by changing only one parameter at a time while the others remain constant. This can reveal the importance of each criterion and its effect on the deforestation related to oil palm. The sensitivity method has been used in various works [49–52].

The proposed approach is based on a conceptual framework and the assessment of a set of criteria related to deforestation. The conceptual framework consists of a set of processes that aims to predict oil palm deforestation.

## 3. Conceptual Framework for Predicting Oil Palm Deforestation

This section proposes a conceptual framework to predict deforestation related to palm oil. The framework involves several processes that should be implemented to detect deforestation: data preparation, creating daily alerts, identifying and assessing criteria, and model training and validation. The processes are shown in Figure 1 below.

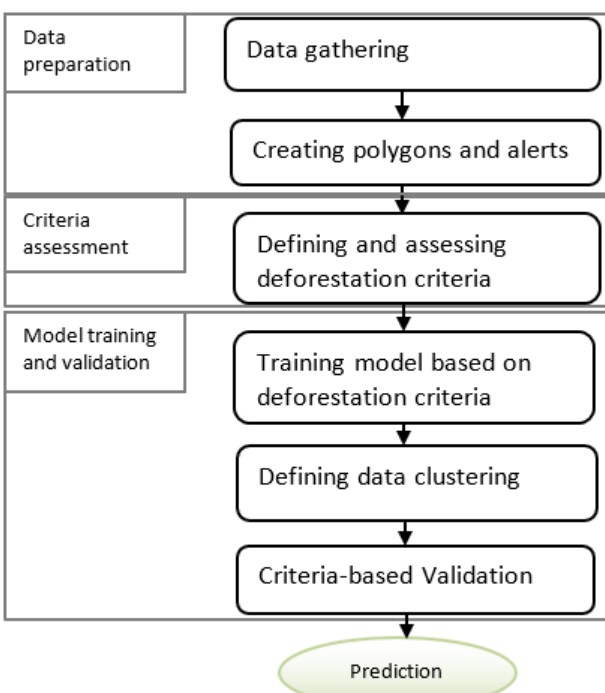

**Figure 1.** Proposed conceptual framework for deforestation prediction.

### 3.1. Data Preparation

We start by preparing data, which is one of the most time-consuming phases in the process of detecting deforestation. Data preparation includes remote sensing image gathering, creating alerts such as polygons, and attaching alerts to polygons.

Creating Daily Polygons and Alerts

In order to identify global deforestation areas, we transform satellite images into daily events that are represented by polygons on the map. Each polygon contains the date of the deforestation detection. Then, we cluster the daily polygons (i.e., patches) based on time and distance. Two or more polygons are merged together if they have the same deforestation day and they are directly adjacent. The clusters are defined based on

thresholds of time and distance. The thresholds are determined based on heuristics learned through past experience of deforestation. We create and attach an alert to each cluster. The creation of an alert on the map aims to make users aware of the deforestation area related to palm oil.

### 3.2. Identifying and Assessing Criteria

In order to interpret deforestation related to oil palm, we defined a set of criteria: distance to oil palm, (min, max, mean), distance to mills (nearest mills' distance), distance to roads, distance to water, distance to oil palm concessions, related alerts, number of patches, distance between patches (mean distance between patches inside the alert), and alert duration, as shown in Figure 2.

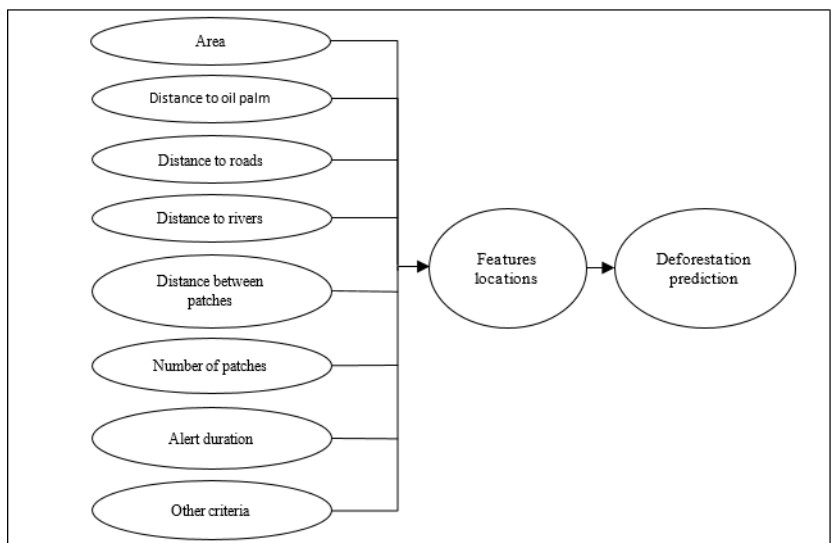

**Figure 2.** Defined criteria for predicting deforestation related to oil palm.

As for the perimeter, area, slope, and elevation criteria, they have been used in several studies. We will not discuss them in detail as they are widely covered in previous studies [53,54]. Figure 2 shows a set of criteria used to interpret deforestation related to oil palm.

1.  Area: This criterion helps us to determine if a given deforestation area is related or not to oil palm, as it is likely for farmers to cut a minimum of 0.5 hectare to establish new oil palm plantations.
2.  Distance to oil palm: We defined this criterion to determine how far oil palm plantations are from the deforestation events. The general idea is that it is more likely for farmers to use the deforested patches for plantations that are close to their already-existing plantations.
3.  Distance to mills and roads: This criterion is used to determine how far the deforestation event is from the nearest mill or road. In fact, it is likely for farmers to make roads inside their plantations to help them collect oil palm easier and faster as such palm needs to arrive fresh in the mill (generally before 24 h from harvesting).
4.  Distance to water: This criterion indicates that it is likely for farmers to use the deforested patches close to water for planting oil palm trees. This is the case especially for smallholders as oil palm requires a lot of water for irrigation, especially in the dry season [55]. In addition, making water pipelines for irrigation is very expensive.
5.  Number of patches: The number of deforestation patches inside a given area should be assessed in order to control palm oil deforestation. This number should not exceed a certain limit.

6. Distance between patches: This criterion indicates that it is more likely for deforested patches close to each other in a given area to turn into oil palm plantations.
7. Alert duration: This criterion calculates the duration of the alert in days between the start date and the end date of the detection.

Criteria are characterized by rates that reflect their spatial variability and weights expressing their influence on deforestation.

We should notice that this restricted set of criteria assessments does not aim to be complete. A restricted set could allow us to predict deforestation related to palm oil. Future implementations should include other criteria, such as plantations for other crops, logging, and urbanization.

### 3.3. Training and Validation

Training and validation data aim to predict new data. We train the model to predict whether or not recent deforestation is caused by oil palm plantations.

We used sensitivity analysis via multiple iterations with different subdatasets to identify the best possible combinations of criteria with their corresponding thresholds.

The following section describes the implementation of the aforementioned processes of the conceptual framework, and presents the implementation results.

### 4. Implementation and Results

In order to show how our proposed framework works, we gathered images from the Sentinel satellites of the European Space Agency (ESA) and the Landsat satellites from NASA. Then, we transformed these images into useful information that can be used to predict deforestation. This transformation includes atmospheric correction and cloud masking.

### 4.1. Study Area

The area that was chosen for implementing our proposed framework is the state of Aceh, located in the north of Sumatra (Indonesia). The study area is depicted in Figure 3, which shows the state of Aceh, which is known for severe deforestation and a high oil palm plantation density and a lot of oil palm concessions. Indonesia was known to have the highest deforestation rate in the world.

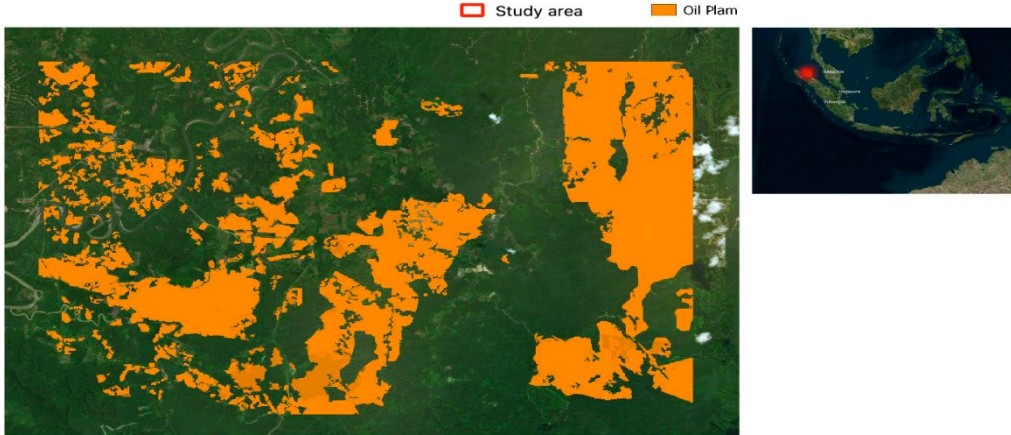

**Figure 3.** Study area: the state of Aceh in Indonesia.

We gathered data related to deforestation events between 1988 and 2020, a land cover map, Sentinel-1 Google Earth and Bing aerial imagery, maps of yearly distances to oil palm plantations from 1988 to 2020, a list of oil palm concessions, and a list of roads.

We used the well-known normalized difference vegetation index (NDVI) to define deforested areas. The NDVI is an indicator of vegetation health, because degradation of

ecosystem vegetation, or a decrease in green, would be reflected in a decrease in NDVI value. The decrease in NDVI values may indicate a degradation of vegetation cover (i.e., deforestation) [56].

We obtained a raster map where each pixel depicts the day and year of deforestation. The darker pixels represent more recent deforestation events, while the brighter pixels depict less recent deforestation events. The resulting raster map can be seen in Figure 4.

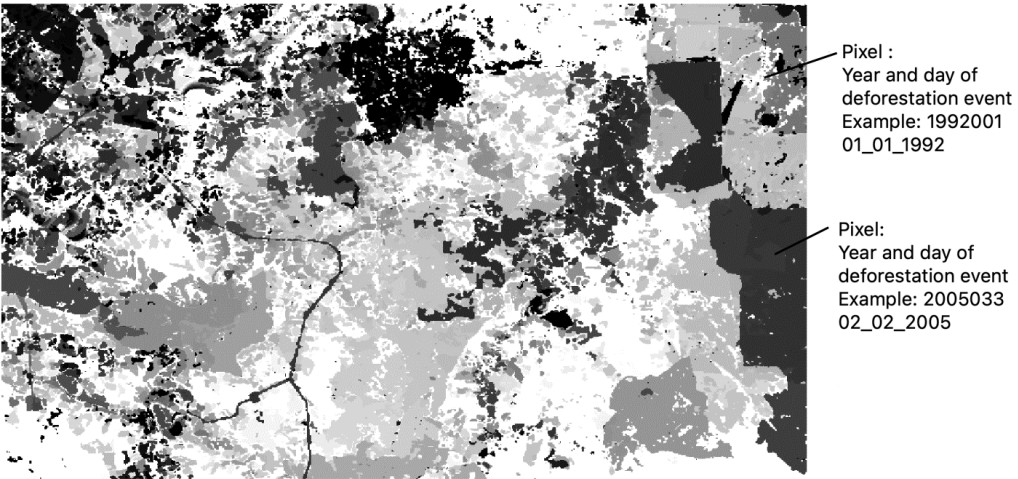

**Figure 4.** Deforestation of North Sumatra, Indonesia, 1988–2021.

The map above shows the deforestation dates pixel by pixel, which were then grouped together if they are directly adjacent or they have the same day of deforestation.

We then created a deforestation and water map by extracting oil palm and water pixels from the land cover map. Water and oil palm features can be seen in Figure 5. We implemented the extraction process using numpy and rasterio Python libraries. Python libraries are very useful for carrying out image processing to extract or enhance information useful for mapping purposes.

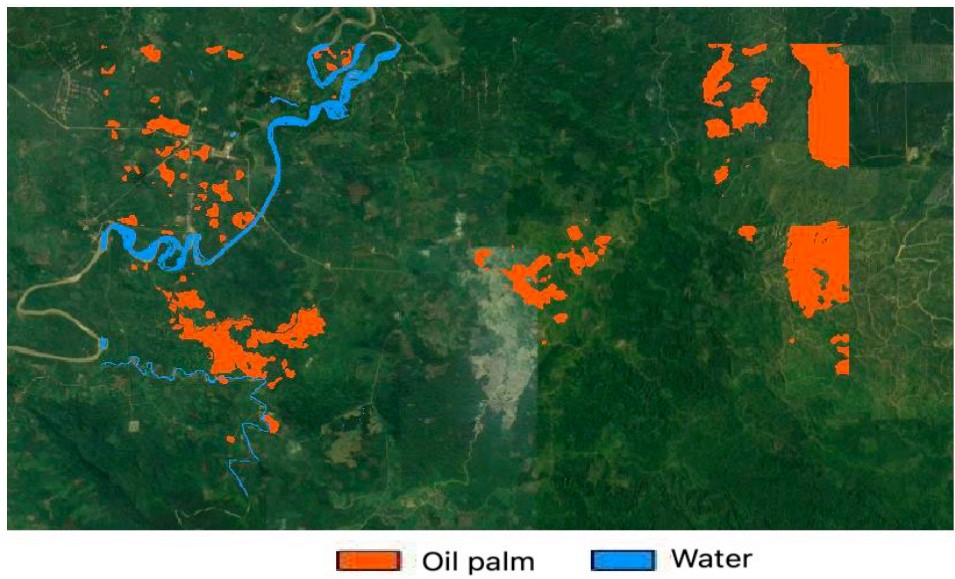

**Figure 5.** Map of oil palm and water grabbed from the classification map.

### 4.2. Creation of Daily Polygons and Alerts

We create daily polygons, which consist of transforming the deforestation map from raster into vector (polygons). Figure 6 shows the obtained polygons that represent defor-

estation. This transformation allows us to define each deforestation patch as a polygon instead of pixels. The reason for the vectorization (transforming raster to vector) is to use the area that each deforestation patch occupies as an independent feature that can be easily described by a set of attributes. Based on polygon features, we can clearly define the boundary of the deforestation patch. To create daily polygons, we developed a Python using rasterio, shapely, and geopandas libraries.

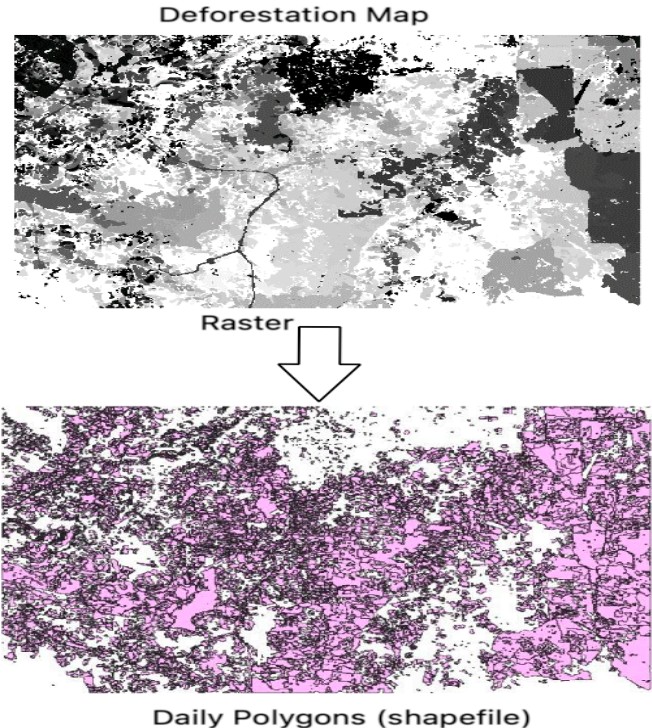

**Figure 6.** Creating daily polygons.

We then create alerts by clustering daily polygons based on time and distance. Alerts are basically multipolygons; polygons are grouped together if they are within 45 days and the distances between patches are not more than 250 m. We set the threshold as 45 days for the time and 250 m for the distance based on heuristic analysis. Alerts and polygons are shown in Figure 7. The highlighted polygon on the left image depicts a daily polygon detected on 7 July 2010, while the highlighted multipolygon on the right side depicts an alert (multipolygon) from 7 to 25 July 2010.

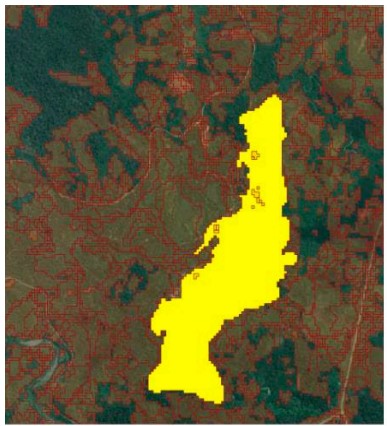
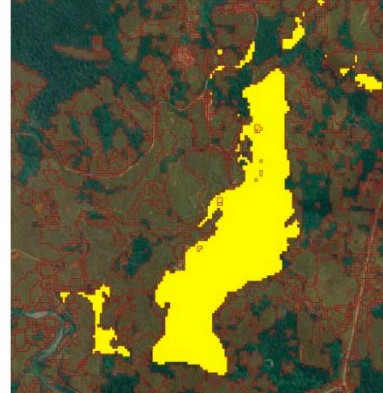

**Figure 7.** Daily Polygons (**left**) and Alerts (**right**).

### 4.3. Implementation of Criteria Assessment

We developed a Python program that calculates areas and yearly distances criteria (distance to oil palm, distance to roads, and distance to water) based on the information extracted from a deforestation oil palm map. We used Python as it is an excellent language for implementing data analysis [26,57,58].

The area and perimeter are calculated using the area and length methods of the geopandas library. With regard to distance between patches, it is calculated based on their mean distance inside the same alert.

In order to assess the distance criteria, we defined a set of assessment methods, which are:

- all;
- only_mean;
- only_max;
- only_min.

We calculated the yearly distances for each year between 1988 and 2020.

Table 1 represents assessment methods used to evaluate the distance criteria.

**Table 1.** Assessment methods for the evaluation of distance criteria.

| Distance to Oil Palm | Distance to River | Distance to Road |
|---|---|---|
| distance_op_mean | distance_rivers_mean | distance_road_mean |
| distance_op_min | distance_rivers_min | distance_road_min |
| distance_op_max | distance_rivers_max | distance_road_max |
| **Distance to Vector** | | |
| distance_concession | | |
| distance_between_patches | | |

The model will be trained based on all these assessment methods; then the best method will be selected, the one that leads to the best prediction accuracy.

### 4.4. Training and Validation

#### 4.4.1. Model Training

The model is trained using the random forest model, which has been very successful as a general-purpose classification and regression method. We chose the random forest algorithm as it can handle vast amounts of data, is less prone to overfitting, and can be analyzed with less time complexity [59].

For the model to be effectively trained, we need to define thresholds for image classification and feature detection [60,61]. In our case study, the land cover threshold has been defined based on heuristic experiences. In this study case, the threshold is defined for both the oil palm area and the distance between patches. For the area, if the alert contains 30% oil palm and contains at least 1 ha oil palm, then it could be considered deforestation related to oil palm, or else it will be considered deforestation related to other drivers. With regard to the distance between patches, only patches within distance mean superior or equal to 300 m will be clustered together. Figure 8 shows an example of an alert represented on the raster map.

#### 4.4.2. Enhancing Accuracy

In order to enhance the optimum structure of the model and its accuracy, we perform hyperparameter tuning. Performing hyperparameter tuning can minimize loss function and give good results in the classification. For instance, Soomro et al. defined hyperparameters for three machine learning models (ANN, SVM, and KNN) to enhance the result of predicting the performance of the stratified thermal energy storage tank [62].

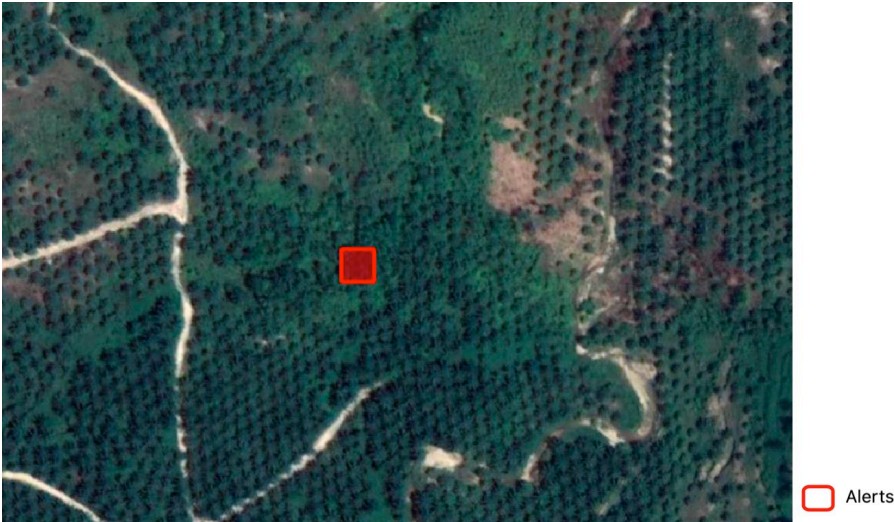

**Figure 8.** Alert inside the oil palm plantation.

We used hyperparameter tuning to identify the best-fitting parameters among the ones given to the model. Therefore, to determine the best-fitting parameter, we define parameters and their corresponding ranges. Our model tuning is based on 35 samplings of hyperparameter values. Table 2 shows the range of the parameters given to the model.

**Table 2.** Parameters and ranges used in hyperparameter tuning.

| Parameter | Range | Explanation |
| --- | --- | --- |
| max_depth | 10–100, None | Maximum depth in each decision tree |
| n_estimators | 10–1000 | Number of trees |
| min_samples_split | $10^{-5}$–$5 \times 10^{-1}$ | Minimum samples in each node before the node is split (fraction) |
| min_samples_leaf | $10^{-5}$–$5 \times 10^{-1}$ | Minimum number of data points in a leaf node (fraction) |
| max_features | auto, sqrt | Maximum number of features considered when splitting a node |
| bootstrap | true, false | Random set of data for each tree |

This tuning was performed with the 'mean' dataset. The highest accurate model is performed with a threshold for oil palm $\geq$30% and palm oil area $\geq$1 ha.

4.4.3. Sensitivity Analysis

In order to select parameters and define an order of importance for these parameters, we study the impact of each parameter on the accuracy of the model. For that, we fix all parameters except one; then we train multiple models with different values of that parameter. The higher is the magnitude of change in input parameter sensitivity, the higher is the importance of the parameter. Sensitivity analysis is a popular feature selection approach employed to identify the important features in a dataset. In sensitivity analysis, each input parameter is selected one at a time, and the response of the machine learning model is examined to determine the parameter's importance rank. Similar methods have been used to determine the optimum weights of parameters for the training model [52,63,64].

Figure 9 shows the model accuracy obtained when changing only one parameter at a time while keeping the others constant.

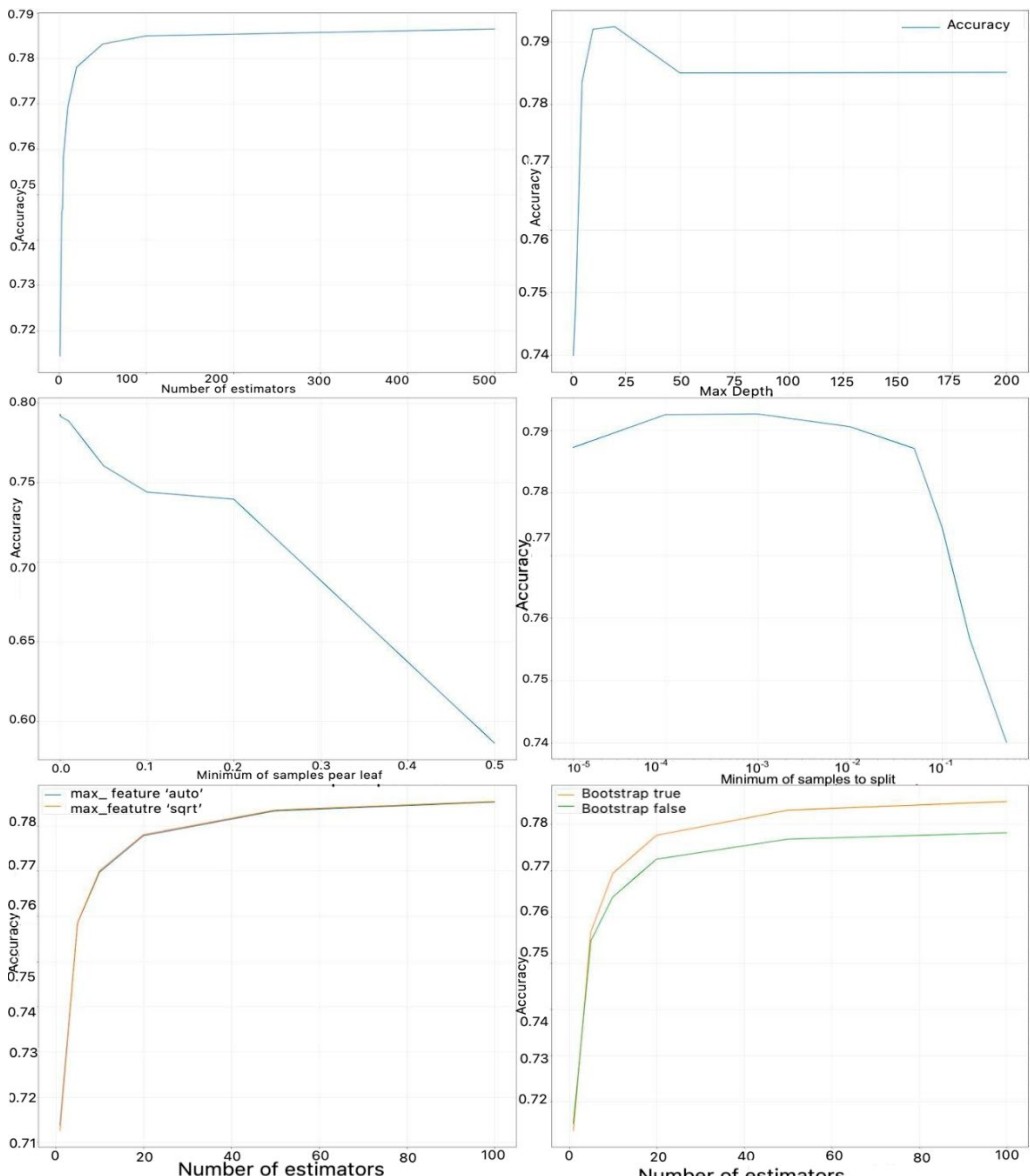

**Figure 9.** Changing one parameter while keeping the others fixed.

We also deal with the change of the clustering period and its impacts on the precision of the model. Therefore, to check whether the clustering period changes the precision of the model or not, three different dates with the same criterion were made as tests to check: 14, 45, and 140 days.

The graph represented in Figure 10 shows that the worst accuracy was in clustering patches within 45 days. The clustering within 140 days is quite good, but the best one is clustering within 14 days, where the less clustering period, the less avoiding multidrivers.

The best model was achieved based on the 'all' assessment method (see Section 4.3) with an accuracy of 82% in total area. This was performed with the following parameters:

— max_depth = 10;
— n_estimators = 100;
— min_samples_split = $10^{-4}$;
— min_samples_leaf = $10^{-5}$;
— max_features = 'auto';
— bootstrap = true;
— clustered within 14 days.

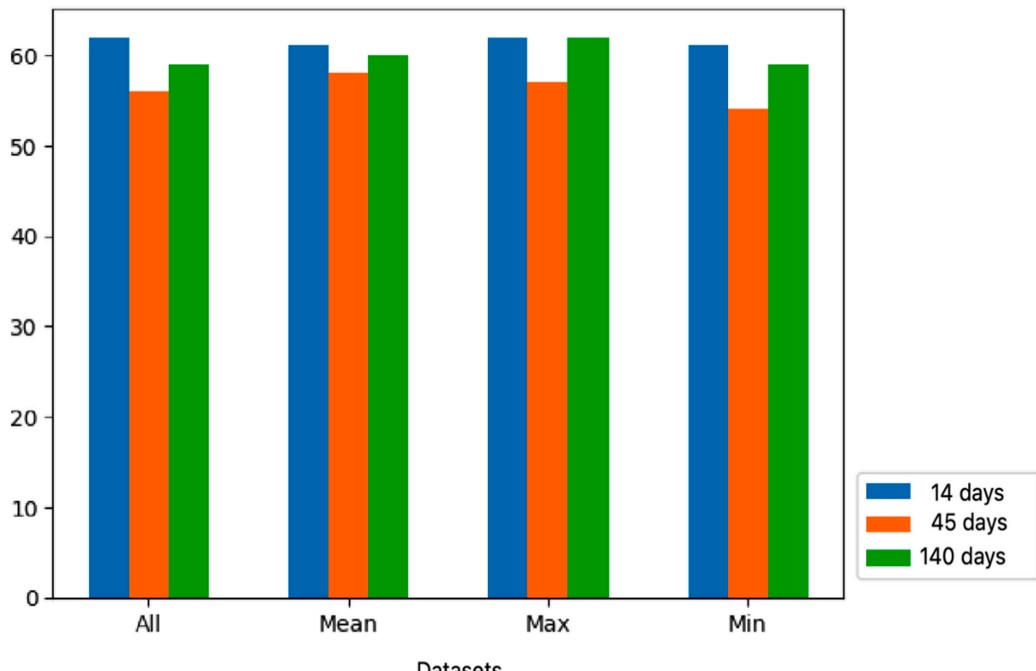

**Figure 10.** Accuracy vs. clustering thresholds.

When you consider the percentage and the area of the palm, the entire dataset reveals a lot of correlation between its criteria and the threshold of 0. The second-best dataset is with a threshold value of 30% and this for percentage and palm area >=1 ha.

Table 3 show the results of our training model for deforestation prediction.

**Table 3.** Results of our training model for deforestation prediction.

|  | **Precision** | **Recall** | **F1 Score** |
|---|---|---|---|
| 1 (other) | 0.97 | 0.98 | 0.97 |
| 0 (oil palm) | 0.72 | 0.66 | 0.69 |
| accuracy |  |  | 0.95 |
| macro avg | 0.84 | 0.82 | 0.83 |
| weighted avg | 0.95 | 0.95 | 0.95 |

We measure the performance of our prediction model based on the F1 score (aka F-measure), which is a popular metric for evaluating the performance of a classification model. The F1 score calculation results in a set of different average scores (macro and weighted) in the classification report.

The model showed an accuracy of 0.82 of deforestation. It also indicates an F1 score of 0.69, a precision of 0.72, a recall of 0.66 for deforestation related to oil palm, an F1 score of 0.97, a precision of 0.97, and a recall of 0.98 for deforestation related to other drivers.

Results of these tests of the sensitivity analysis and the changing of the clustering period show that the importance of a criterion is the following:

1. Area (20.2%);
2. Largest_area (19.7%);
3. Perimeter (15.7%);
4. Distance_op_min (12.5%);
5. Distance_rivers_mean (7.89%);
6. Distance_road_max (7.74%);
7. Distances_patches (5.58%);
8. Distance_concession (4.59%).

Evaluating at the average feature values and the impact on the probability of predicting the polygon as oil palm plantations or not resulted in Figure 11. The blue lines depict when the model was correct in its decision, while the red lines depict when the model was wrong in its prediction. The biggest area's alert is more likely to be used for oil palm, and of course, the perimeter is proportional to the area. That is why they both have the same orientation.

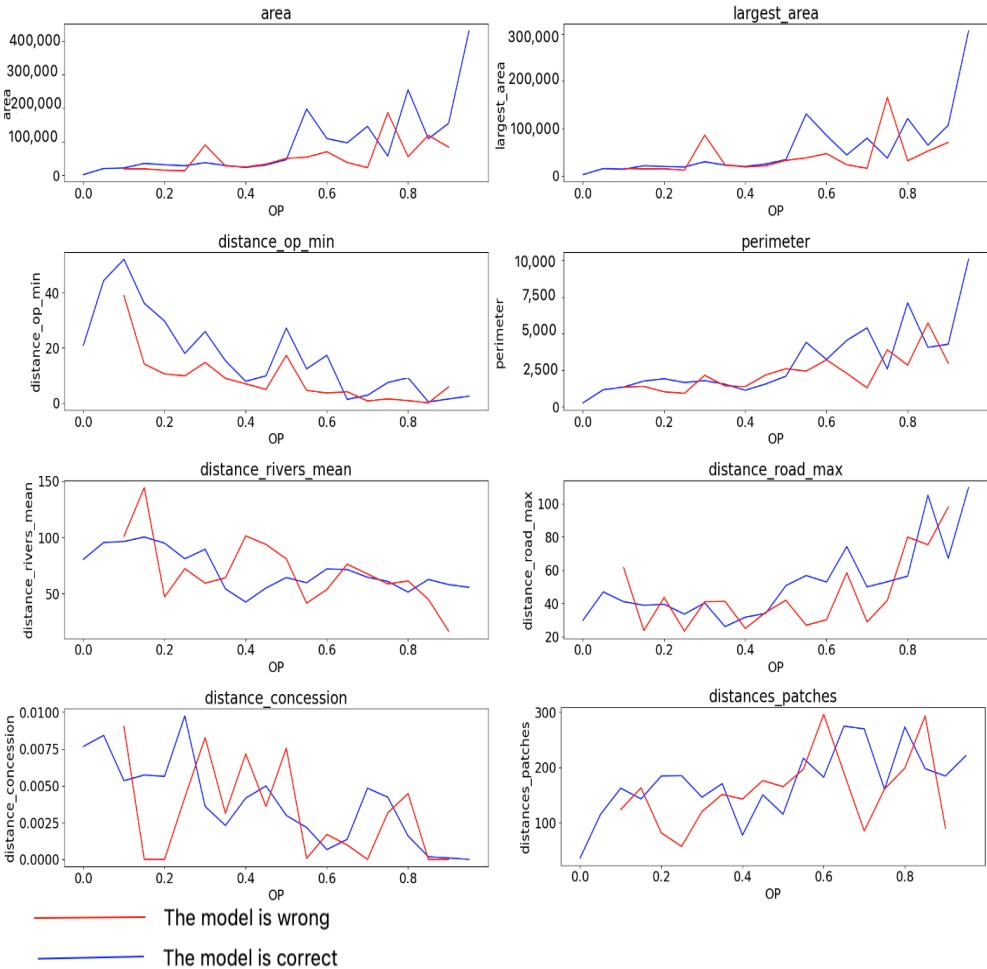

**Figure 11.** Average values of features versus the probability of being labeled as oil palm.

This can also be seen in the largest area metric, where the higher the value is, the more likely it will be used for oil palm plantations.

The closer the polygon is to already-existing plantations, the more likely it is used for oil palm. This can also be seen in the distance to the water. However, inversely for distance to roads, it is expected for patches that are in a way far from the principal roads (i.e., countryside) to be used for oil palm. Additionally, for sure it is predictable for alerts that are close or inside concessions to be used for oil palm.

Figure 11 depicts the average values of features versus the probability of being labeled as oil palm. The blue color indicates that the prediction of the model was correct, and the red color indicates that the prediction of the model was wrong, as shown in Figure 11.

## 5. Conclusions

The work presented in this paper focuses on predicting deforestation related to oil palm, an issue that has not been explored in existing research studies. The work proposes an approach that aims to enhance the prediction accuracy of deforestation related to oil palm. The approach consists of a framework for model training and a set of criteria: area,



distance to oil palm, distance to mills, distance to roads, distance to water, distance to oil palm concessions, related alerts, number of patches, and distance between patches.

The model is implemented using data about forest and oil palm plantations in the Aceh province of Indonesia, which faces one of the highest deforestations in the world.

The model implementation showed an acceptable accuracy of 0.82. It also indicates an F1 score of 0.69, a precision of 0.72, and a recall of 0.66 for deforestation related to oil palm and an F1 score of 0.97, a precision of 0.97, and a recall of 0.98 for deforestation related to other drivers.

The most important criterion is the area; then comes the proximity to oil palm plantations, the distance to water, and the distance to roads. The distance to water criterion is also significant and could help to make a distinction between oil palm and other crops.

We should note that this set of criteria assessments does not aim to be complete, but rather to allow us to predict deforestation related to palm oil. Future implementations should include other criteria, such as plantations for other crops, logging, and urbanization.

The proposed approach can support decision makers to define strategies concerning deforestation related to palm oil and the implementation of a sustainable forest. Providing information about deforestation prediction can help authorities to make appropriate decisions about where and when they can establish new plantations to ensure a sustainable oil palm. Additionally, based on the deforestation prediction, the companies of palm oil production can quickly determine which parts of their supply chain are not in line with their principles with regard to the sustainability of forests.

Although the model shows an acceptable accuracy for predicting deforestation related to oil palm, there are still some limitations.

- First, relatively large clustering periods (e.g., 14, 45, or 140 days) may affect the performance of the model as it may lead to considering several criteria at the same time.
- Second, larger datasets are needed to accurately train the model and more clearly distinguish oil palm from other plantations.

Further research using larger datasets will result in better prediction of deforestation related to oil palm. In addition, for future work, it is highly recommended to reduce the clustering period in order to avoid considering several criteria at the same time.

**Author Contributions:** Conceptualization, T.S.; methodology, T.S. and S.S.; software, A.L. and T.S.; validation, T.S., S.S. and A.L.; formal analysis, T.S. and S.S.; investigation, T.S. and A.L.; resources, T.S. and A.L.; data curation, A.L.; writing—original draft preparation, T.S., S.S. and A.L.; writing—review and editing, T.S. and S.S.; visualization, A.L., T.S. and S.S.; supervision, T.S.; project administration, T.S. and S.S.; funding acquisition, n/a. All authors have read and agreed to the published version of the manuscript.

**Funding:** This research received no external funding.

**Institutional Review Board Statement:** Not applicable.

**Informed Consent Statement:** Not applicable.

**Acknowledgments:** Many thanks to the stuff of Satelligence who helped with of this work. Big thanks to Rens Masselink, Loes Masselink, Michel Wolters, Berto Booijink, Vincent Schut, and Rob Verhoeven.

**Conflicts of Interest:** The authors declare no conflict of interest.

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
