# Peer review of "A Machine-Learning-Based Approach to Predict Deforestation Related to Oil Palm: Conceptual Framework and Experimental Evaluation"

_applsci, doi:10.3390/app13031772_

Round 1

Reviewer 2 Report

1. numbering of citations in the text should be in a sequence not in a zigzag citation of references.

2. connection b/w paragraphs are very poor. be consistent in a literature review in defining the scope, method and existing results. 

3. kindly review the following paper in terms of the arrangement of the machine learning approach.  and cite it in your manuscript;

"Machine Learning Approach to Predict the Performance of a Stratified Thermal Energy Storage Tank at a District Cooling Plant Using Sensor Data."  Sensors 22 (19):7687. doi: doi.org/10.3390/s22197687."

4. cite all tables and figures

5. figure 3 to 9 are not readable and not cited

6. make consistent and connected in between all paragraphs, especially in the conclusion. 

Round 2

Reviewer 2 Report

all comments are incorporated without citation of all figures.

please check citation figures carefully 
